# Effects of Dietary Protein Restriction on Colonic Microbiota of Finishing Pigs

**DOI:** 10.3390/ani13010009

**Published:** 2022-12-20

**Authors:** Shanghang Liu, Zhiyong Fan

**Affiliations:** Animal Nutritional Genome and Germplasm Innovation Research Center, College of Animal Science and Technology, Hunan Agricultural University, Changsha 410128, China

**Keywords:** finishing pigs, gut microbiota, low-protein diets, colonic microflora

## Abstract

**Simple Summary:**

Low-protein diets can effectively alleviate the pressure of protein resource shortage and nitrogen emission from the pig industry. In recent years, the effects of low-protein diet and amino acids or additives on growth performance, meat quality and odor emission of growing finishing pigs have been studied. However, gut health is also an important indicator, as well as whether the changes in intestinal flora caused by the reduction in dietary protein levels are beneficial to the growth of finishing pigs. Additionally, how will the metabolites of the microbes change when the microbes change in the hindgut?

**Abstract:**

This study is aimed at the effects of low-protein diets with four amino acids balanced on serum biochemical parameters and colonic microflora of finishing pigs. Fifty-four healthy (Duroc × Landrace × Yorkshire) hybrid barrows with an average body weight of 70.12 ± 4.03 kg were randomly assigned to one of three dietary treatments with three barrows per pen and six pens per treatment. The barrows were fed a normal protein diet (NP), a low-protein diet (LP), and a very low-protein diet (VLP). Compared with the NP diet, reduced dietary protein did not influence serum biochemical parameters (*p* > 0.05). The valeric acid was significantly increased with the VLP diet (*p* < 0.05). Compared with the NP diets, the abundance of Terrisporobacter (13.37%) Clostridium_sensu_stricto_1 (23.37%) and Turicibacter (2.57%) increased to 21.04, 33.42 and 13.68% in LP diets and 16.72, 43.71 and 14.61% in VLP diets, while the abundance of Lactobacillus (9.30%) and Streptococcus (25.26%) decreased to 3.57 and 14.50% in LP diets and 1.86 and 4.07% in VLP diets. Turicibacter and Clostridium_sensu_stricto_6 had a powerful negative correlation with the content of valeric acid (*p* < 0.01), while Peptococcus and Clostridia_UCG-014 had a very solid positive correlation (*p* < 0.01). In conclusion, reducing dietary protein level can improve colon microbiota composition, especially reducing the abundance of bacteria related to nitrogen metabolism, but has no significant effect on SCFA except valeric acid. In addition, reduction in the dietary protein level by 5.48% had more different flora than that of 2.74% reduction in dietary CP level.

## 1. Introduction

Dietary protein is the fundamental source of amino acids for livestock. However, simply increasing the protein content of the diet is not necessarily beneficial to the growth of pigs but will increase the nitrogen emission and waste of protein raw materials in production. The shortage of protein source and the nitrogen excretion environmental pollution is a serious global problem at present. The need of animals for protein is essentially the need for amino acids [1], so we can ensure the supply of amino acids to animals while reducing the level of dietary protein, so as to ensure that the performance of livestock and poultry is not reduced. Previous studies reported that low-protein diets could decrease the concentration of ammonia–nitrogen in feces and urine of finishing pigs [2,3], but the effects on gut health have not been well-documented.

Gut microbiota represents a large and complex microbial community composed of at least 500 to 1000 species in mammals [4]. Gut microbiota communities interact with each other and their host, which plays a vital role in the host physiology and metabolism, including modulation of energy harvest, nutrient metabolism and immune system development [5]. Protein is one of the most common and principle components in diets, the portion of the dietary nitrogenous compounds escape digestion in the small intestine and enter the large intestine to be further utilized by the hindgut microbiota (mainly at the distal colon) [6]. Then, these dietary nutrients are fermented by microorganisms to produce metabolites such as short-chain fatty acids (SCFAs) and biogenic amines [7].

The bacterial community balance in the gut of finishing pigs has been established, and the bacterial composition structure remains relatively stable [8]. However, even after climax communities are already established, microbial composition changes dynamically in response to new microbial colonization, inflammatory stress and diet [9]. Therefore, the current study aimed to evaluate the effect of low-protein diets balanced with amino acid supplemented on gut microbes in pigs and to investigate the association between colonic SCFAs and gut microbes under low-protein diets.

## 2. Materials and Methods

The study protocol was approved by College of Animal Science and Technology, Hunan Agricultural University, and the treatment and slaughtering conditions were in accordance with the Animal Care and Use Guidelines of China (The ethical code: ACC20220556).

### 2.1. Animal Treatment and Experimental Design

Fifty-four healthy (Duroc × Landrace × Yorkshire) hybrid barrows with an average body weight of 70.12 ± 4.03 kg were randomly assigned to one of three dietary treatments with three barrows per pen and six pens per treatment. The barrows were fed a normal protein diet (NP), a low-protein diet (LP), and a very low-protein diet (VLP). As shown in Table 1, all the diets were balanced with four essential amino acids (lysine, methionine, threonine and tryptophan) and crude protein contents were 13.50% (NP; CON), 10.76% (LP) and 8.02% (VLP). The experiment lasted 37 days, during which all the pigs were given free access to clean drinking water and the diet assigned to them.

The animal feeding experiment was conducted in Liuyang Animal Testing Base in Hunan Province. The experiment period was 37 days, and the pigs were raised in the whole leaky seam floor piggery, free to eat and drink according to the routine procedures of deinsectization and immunization and regular cleaning and disinfection of piggery. The thermostat automatically regulates the ambient temperature in the house and the window is opened for ventilation at regular times. Pigs are not fed on the morning of feeding day 38; one pig was randomly chosen from each pen (*n* = 6) and selected for slaughter, blood was collected before slaughter and serum was separated by centrifugation and placed at −20 °C. Colonic digesta was collected and stored at −80 °C.

### 2.2. Direction Indicators

#### 2.2.1. Blood Biochemical Parameters

Blood samples were harvested from anterior vena cava and serum were separated after centrifugation at 300 rpm for 10 min under 4 °C. Cobas c-311 coulter chemistry analyzer (Roche Diagnostics International Ltd., Shanghai, China) was used to test serum biochemical parameters, including total bile acid (TBA), triglyceride (TG), total cholesterol (TC), low-density lipoprotein (LDL), high-density lipoprotein (HDL) and glucose (Glu).

#### 2.2.2. Short-Chain Fatty Acids Analysis

To determine short-chain fatty acid content in colonic chyme, refer to Yu et al. [10]. In brief, 1.5 g ileal or colonic digesta was suspended in 1.5 mL of distilled water and centrifuged at 15,000× *g* at 4 °C for 10 min. One milliliter of supernatant mixed with 200 μL of metaphosphoric acid was put in an ampoule and set in an ice bath for 30 min and centrifuged for 10 min. Samples were inserted into an HP 6890 Series Gas Chromatograph (Hewlett Packard, PA, California, CA, USA) with an HP 19091N-213 column with 30.0 m × 0.32 mm i.d. (Agilent, PA, California, CA, USA). Temperatures for injector and detector were set at 185 °C and 210 °C, respectively. Each sample was measured three times.

#### 2.2.3. Microbiota Analysis by 16S RNA

Total genome DNA was extracted from colon samples from growing pigs using the QIAamp Fast DNA Stool mini kit (Qiagen, Hilden, Germany) and checked with 1% agarose gel. The DNA concentration and purity were determined with Nano Drop 2000 UV-vis spectrophotometer (Thermo Fisher Scientific, Wilmington, NC, USA). The specific primer with the barcode (16S V3-V4) was amplified by an ABI Gene Amp R9700 PCR thermocycler (ABI, Los Angeles, CA, USA). Then, the PCR products were extracted, purified and quantified. Paired-end sequencing was performed on an Illumina MiSeq PE300 platform/NovaSeq PE250 platform (Illumina, San Diego, CA, USA). The raw 16S rRNA gene sequencing reads were demultiplexed, quality-filtered and merged according to previous studies [11,12]. The complexity of species diversity was evaluated with ACE and Chao richness estimators and diversity indices of Shannon and Simpson [13]. β-diversity was evaluated using principal component analysis (PCA) based on the Euclid distance. The significant differences between samples were evaluated by the analysis of similarities (ANOSIM). OTUs representing < 0.005% of the population were removed, and taxonomy was assigned using the RDP classifier. The relative abundance of each OTU was counted at different taxonomic levels. Then, bioinformatics analysis was mainly performed using QIIME (V1.7.0; San Diego, CA, USA) and R packages (Version 3.3.1, R Core Team, Vienna, Austria). The OTU table in QIIME was used to calculate OTU-level, and β-diversity was assessed by principal coordinate analysis (PCoA). The cluster analysis and significant differences between samples were tested by ANOSIM [14].

### 2.3. Statistical Analysis

Data for blood variables and SCFAs were subjected to analysis of variance (ANOVA) suited for a randomized complete block design using the general linear model (GLM) procedure (version 9.2, SAS Institute, Inc., Cary, NC, USA.). For the growth performance, pen served as the experimental unit. Results are expressed as the mean + standard error of the mean (SEM). Statistical differences among groups were separated by the Bonferroni multiple comparisons test. *p* values of < 0.05 were significant for all data in this paper.

## 3. Result

### 3.1. Serum Biochemical Parameters

The effect of dietary crude protein level on the serum biochemical parameters of finishing pigs is presented in Table 2. Reduced dietary protein did not influence serum biochemical parameters (*p* > 0.05).

### 3.2. SCFAs

Table 3 shows that different protein diets can significantly affect the content of short-chain fatty acids in the colon and cecum of pigs. Compared with the NP diet, the valeric acid was significantly increased with the VLP diet (*p* < 0.05). However, other short-chain fatty acids were not significantly different.

### 3.3. Structural Changes in The Microbial Community

Following Illumina MiSeq sequencing analysis, a Venn diagram of the OTUS exhibited that 878, 789 and 761 OTUs were identified, respectively, in the NP, LP and the VLP diets; 620 OTUs were shared by each of the three groups (Figure 1A). Additionally, no remarkable differences were found in the diversity indices (Shannon and Simpson index) (*p* > 0.05), same for the richness estimators (ACE and Chao index) of the colon microbiota (Figure 2). The principal coordinate analysis (PCoA) was used to characterize the β diversity of bacterial communities in the fecal samples of growing pigs in the NP, LP and VLP diets. As shown in Figure 1B, PCoA results present that the colon microbial composition of finishing pigs exposed to low-protein-level groups was distinguishable from that of the NP diets.

As shown in Figure 3, at the genus level, six phyla were detected in three groups of samples. Different colors in the figure represent different bacterial communities. The higher the column, the larger the proportion of the sample and the higher the relative abundance. The relative abundance and proportion of each group at the genus level could be intuitively seen from the species annotation results. It was uncovered that the dominant strains of colon flora of finishing pigs are primarily composed of *Turicibacter*, *Terrisporobacter*, *Clostridium_sensu_stricto_1*, *Lactobacillus*, *Streptococcus*, *UCG-005* and so on. Compared with the NP diets, the abundance of *Terrisporobacter* (13.37%), *Clostridium_sensu_stricto_1* (23.37%) and *Turicibacter* (2.57%) increased to 21.04, 33.42 and 13.68% in the LP diets and 16.72, 43.71 and 14.61% in the VLP diets, while the abundance of *Lactobacillus* (9.30%) and *Streptococcus* (25.26%) decreased to 3.57 and 14.50% in the LP diets and 1.86 and 4.07% in the VLP diets.

As shown in Figure 4, in addition, it was revealed, through LEfSe analysis (LDA threshold: 3.0), that the LP diet was not found to be significant; the VLP diet induced a significant enrichment of *Clostridium_sensu_stricto_1*, *Turicibacter* and *Clostridium_sensu_stricto_6.* By contrast, *Streptococcus*, *Coprococcus*, *Blautia, Rikenellaceae_RC9_gut_group, UCG-008, Muribaculaceae, Clostridia_UCG-014, NK4A214_group* and *Peptococcuswere* were the predominant species in the NP diet.

### 3.4. Correlation of Bacteria and Colonic SCFAs

The correlation between the top 12 genera and the concentration of SCFAs content in colon is shown in Figure 5, wherein the bacteria including *Turicibacter, Clostridium_sensu_stricto_6, Blautia, UCG-008 Clostridium_sensu_stricto_1, Peptococcus, Coprococcus, NK4A214_group, Streptococcus, Muribaculaceae, Rikenellaceae_RC9_gut_group* and *Clostridia_UCG-014*. *Turicibacter* and *Clostridium_sensu_stricto_6* had a very significant negative correlation with the content of valeric acid (*p* < 0.01), while *Peptococcus* and *Clostridia_UCG-014* had a very solid positive correlation (*p* < 0.01).

## 4. Discussion

Many experimental studies have shown that the reduction in dietary protein level may affect the lipid metabolism status of animals [15,16] because the use of low-protein diets can reduce animal energy loss, protein turnover and animal heat production, so that energy use efficiency in the body increases [17,18]. However, we did not observe significant changes in lipid levels. It is worth noting that the contents of TG, TC and HDL-C in the VLP diet tend to increase compared with the control group. According to Madeira et al.’s [19] study, the restriction of dietary protein increased total lipids, total cholesterol, high-density lipoprotein cholesterol and low-density lipoprotein cholesterol. Additionally, the reason we did not see a significant change probably has to do with how long the pigs were fasting.

Intestinal microbiota and its fermentative metabolites have potential effects on host health. Chao estimate indicated the bacterial richness and Shannon index reflected the bacterial diversity [20]. In our experiment, with four balanced essential amino acids, reducing the dietary CP level had no significant effect on the α diversity index of colonic microbiota in finishing pigs, which was consistent with Zhou et al. [21]. Meanwhile, Fan et al. [22] found that the Chao and Shannon index of the bacterial community in the colon was not significantly different. Though bacterial diversity was slightly affected, the samples of the three groups were clustered separately. Furthermore, a separation between colon samples of the NP and VLP diets could be best observed from PCoA results. Therefore, a reduction in dietary protein concentration was indicated to affect the colonic microbiota.

Species richness of colonic microbiota revealed that the abundance values of *Terrisporobacter, Clostridium_sensu_stricto_1* and *Turicibacter* were increased with reduction in dietary protein. *Turicibacter* is positively correlated with colitis [23]. *Clostridium_sensu_stricto_1* is an opportunistic pathogen [22,24], which can cause intestinal inflammation and decrease the content of SCFAs [25]. However, we did not observe significant reduction in SCFAs in colonic contents in our study. On the contrary, the content of valeric acid was significantly increased with the VLP diet. Fan et al. [15] also found the same result in the colon, but no significant damage on colonic morphology and expression of tight junction proteins were observed in low-protein diet groups. It may be partly due to the mild change in the colonic bacteria community. The literature also reported that *Terrisporobacter* and *Clostridium_sensu_stricto_1* were fiber-degrading bacteria at each growing stage [26,27,28]. In our study, soybean meal was partially replaced by corn and bran in the preparation of the low-protein diet. Therefore, there were more fibers and starches in the VLP diets, which could increase the abundance of *Terrisporobacter* and *Clostridium_sensu_stricto_1*. The carbohydrates which are available to microorganisms are rich in variety and complex in structure, including resistant starch, non-starch polysaccharide and indigestible monosaccharides [29]. Reduced dietary protein levels relatively reduce abnormal protein fermentation in the hindgut, while short-chain fatty acids produced by carbohydrate fermentation are rapidly absorbed in the hindgut, providing approximately 30% of the energy of pigs [30].

Gut microbiota can synthesize abundant proteases and peptidases to hydrolyzed proteins and peptides and catabolize almost all kinds of amino acids (AA) [31]. *Clostridia, streptococci, Fusobacterium* and *Lactobacillus* are also frequently seen proteolytic bacteria [32,33]. At the same time, *Lactobacillus* and *Streptococcus* are capable of producing amines through the decarboxylation of AA. In this study, we observed a decrease in the abundance of Streptococcus in the colon of VLP diets. It is worth noting that the proportion of *RC9_gut_group* belongings to *Rikenellaceae*, *Lachnospiraceae* belonging to *Firmicutes*, *Coprococcus* and 12 other bacterial genera were decreased in the LP diet. The decrease in abundance of *RC9_gut_group* may be related to the decrease in dietary nitrogen sources, or the influence of dietary amino acid imbalance on lipid utilization [34]. Interestingly, we found the abundance of *Coprococcus* was significantly decreased in both the NP and VLP diets, which suggested that *Coprococcus* was sensitive to dietary protein changes.

## 5. Conclusions

Reducing dietary protein level can improve colon microbiota composition, especially reducing the abundance of bacteria related to nitrogen metabolism, but it has no significant effect on SCFA except valeric acid. In addition, reduction in the dietary protein level by 5.48% had more different flora than that of a 2.74% reduction in dietary CP level.

## Figures and Tables

**Figure 1 animals-13-00009-f001:**
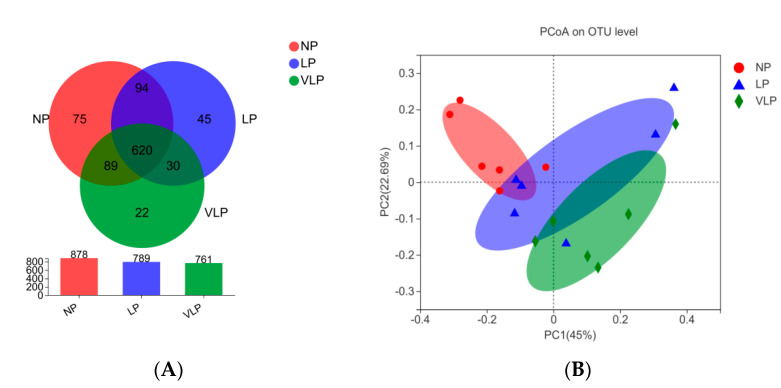
(**A**) Effects of different protein diets on colon microflora OTU of finishing pigs. (**B**) Principal coordinates analysis (PCoA) of microbial composition in the colon of growing pigs (based on the bary_curtis distance). The individual pig was regarded as the experimental unit (*n* = 6). NP, normal protein; LP, low protein; VLP, very low protein.

**Figure 2 animals-13-00009-f002:**
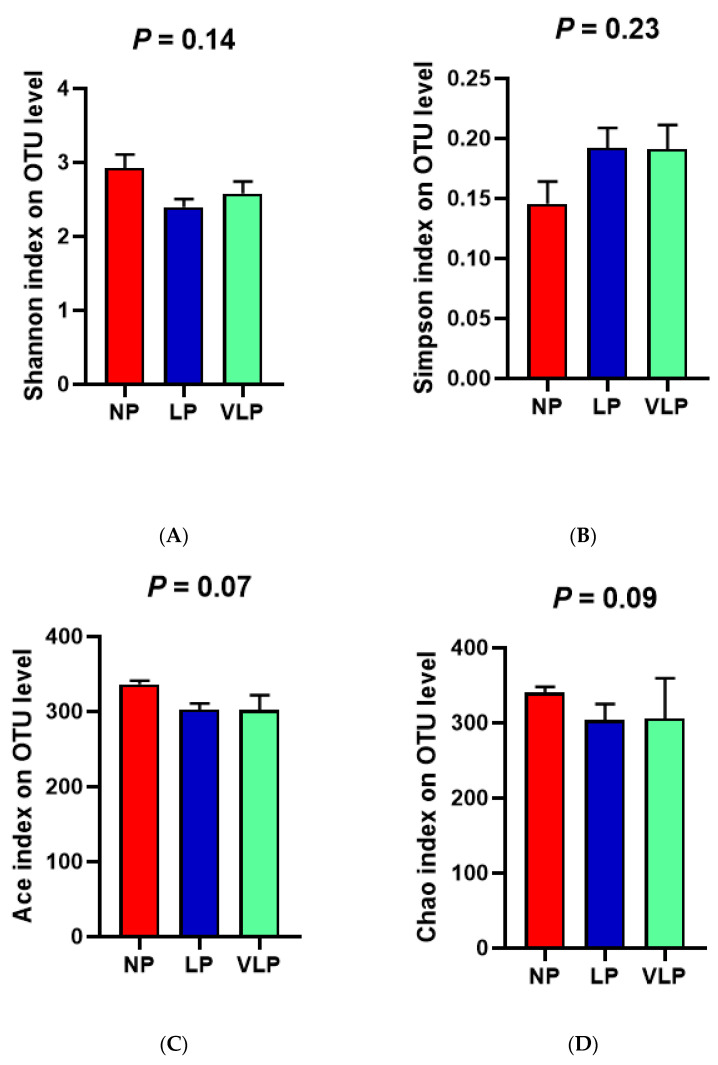
Effects of different protein diets on the alpha diversity of bacterial flora in the colon of finishing pigs. (**A**) Shannon index, (**B**) Simpson index, (**C**) Ace index, (**D**) Chao index. The individual pig was regarded as the experimental unit (*n* = 6). NP, normal protein; LP, low protein; VLP, very low protein.

**Figure 3 animals-13-00009-f003:**
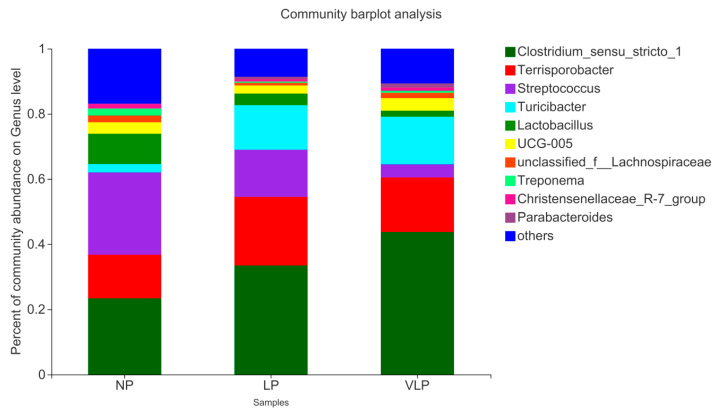
Relative abundance of colonic bacteria based on the genus level. The individual pig was regarded as the experimental unit (*n* = 6).NP, normal protein; LP, low protein; VLP, very low protein.

**Figure 4 animals-13-00009-f004:**
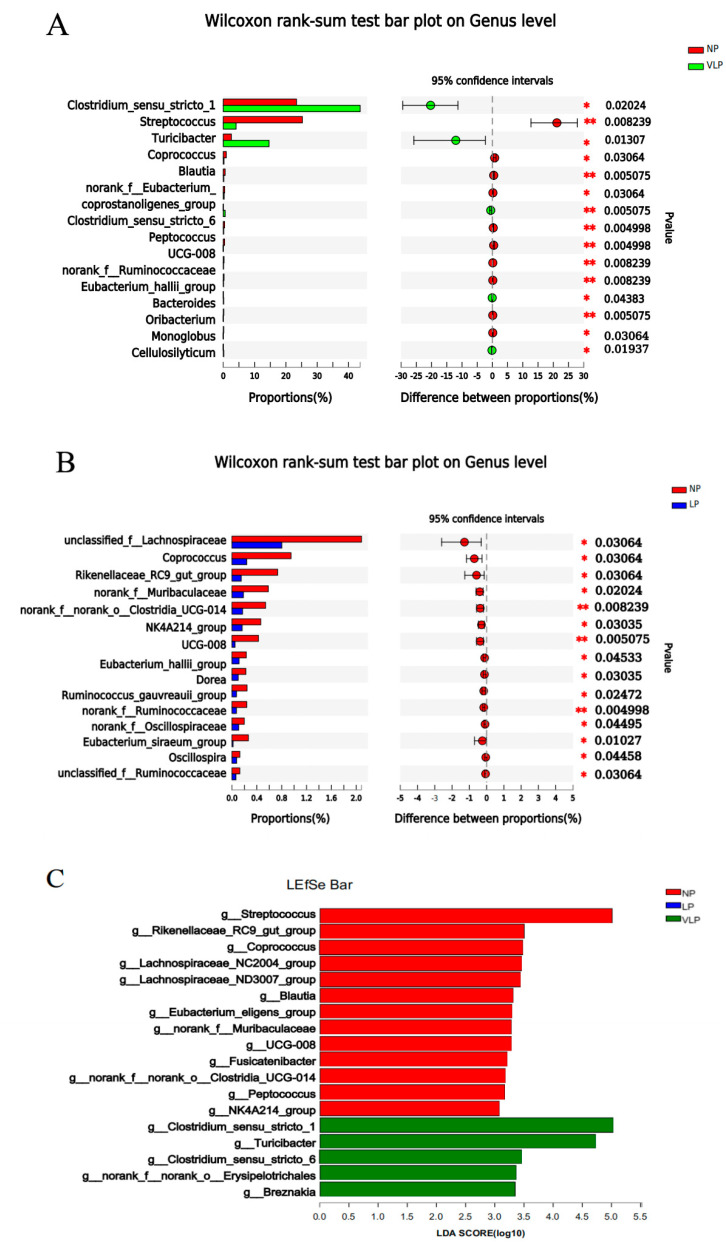
Different microbiota comparison by the student t-test on the genus of the colon is shown in (**A**,**B**). *p*-values (* *p* < 0.05, ** *p* < 0.01) are shown on the right. Identification of the most differentially abundant genera in the colon. The plot (**C**) is generated from Linear Discriminant Analysis Effect Size (LEfSe) analysis with CSS-normalized OTU table and displays taxa with LDA scores above 3.0 and *p*-values below 0.05. Genera enriched in the samples with the NP diet are indicated with red bars, genera enriched in the samples with the LP diet are indicated with blue bars and genera enriched in the samples with the VLP diet are indicated with green bars.

**Figure 5 animals-13-00009-f005:**
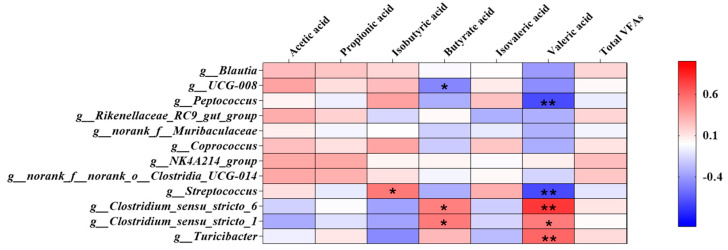
The correlation between the top 12 genera and the concentration of Colon SCFAs content. The individual pig was regarded as the experimental unit (*n* = 6). NP, normal protein; LP, low protein; VLP, very low protein. “*” means there is a significant difference, “**” means a very significant difference.

**Table 1 animals-13-00009-t001:** Ingredient Composition and Nutrient Levels of the Experimental Diets. (%, as-fed basis).

Item	Dietary Treatments ^a^		
	NP	LP	VLP
Corn	78.17	87.38	83.86
Soybean meal	16.56	7.40	-
Soybean oil	1.08	0.05	2.71
Stone powder	1.01	1.08	1.10
Premix ^b^	2.00	2.00	2.00
Mould inhibitor	0.05	0.05	0.05
L-Lysine HCL	0.22	0.48	0.72
Potassium chloride	-	0.25	0.51
Sodium chloride	0.50	0.50	0.50
Calcium hydrogen phosphate	0.37	0.41	0.50
DL-Methionine	0.03	0.10	0.19
Tryptophan	0.01	0.05	0.09
Threonine	-	0.11	0.24
Valine	-	0.07	0.23
Isoleucine	-	0.06	0.21
Histidine	-	0.01	0.10
Phenylalanine	-	-	0.15
Filler	-	-	6.84
Total	100.00	100.00	100.00
Calculated nutrient levels			
Net energy, kcal/kg	2475	2475	2475
Crude protein	13.50	10.76	8.02
Potassium	0.52	0.52	0.52
Calcium	0.52	0.52	0.52
Dispensable amino acids			
SID Lysine	0.73	0.73	0.73
SID Methionine	0.23	0.27	0.31
SID Cystine	0.20	0.16	0.11
SID Methionine + cystine	0.42	0.42	0.42
SID threonine	0.46	0.46	0.46
SID tryptophan	0.13	0.13	0.13

^a^ NP, normal protein; LP, low protein; VLP, very low protein. ^b^ Supplied per kg of diet: vitamin A 18,000 IU, vitamin D 35,000 IU, vitamin E 35 IU, vitamin K 5 mg, vitamin B1 5 mg, vitamin B2 10 mg, vitamin B12 35 μg, iron 66 mg, copper 6 mg, zinc 54 mg, magnesium 15 mg, iodine 0.24 mg, selenium 0.18 mg, niacin 40 mg, pantothenic acid 20 mg, folic acid 1.5 mg.

**Table 2 animals-13-00009-t002:** Effect of Low-protein Diet on blood lipid profiles of Finishing Pigs.(mmol/L).

Items ^1^	NP	LP	VLP	SEM ^2^	*p*-Value
TBA	23.452	20.983	41.785	5.051	0.06
GLU	2.538	5.112	3.393	1.117	0.30
TC	1.593	1.65	1.81	0.135	0.52
TG	0.347	0.367	0.467	0.04	0.13
HDL-C	0.653	0.667	0.797	0.073	0.35
LDL-C	0.735	0.677	0.563	0.075	0.31

^1^ TBA, total bile acid; HDL-C, high-density lipoprotein; LDL-C, low-density lipoprotein. ^2^ SEM, standard error of the mean (*n* = 6). NP, normal protein; LP, low protein; VLP, very low protein.

**Table 3 animals-13-00009-t003:** Effect of Low-protein Diet on SCFAs content in colon of Finishing Pigs (mg/kg).

Items ^1^	NP	LP	VLP	SEM ^2^	*p*-Value
Acetic acid	416.07	382.35	390.74	21.41	0.53
Propionic acid	165.42	160.43	163.01	10.51	0.95
Isobutyric acid	16.03	14.78	13.05	1.23	0.27
Butyric acid	99.43	116.50	129.12	13.76	0.23
Isovaleric acid	25.24	24.55	24.54	2.51	0.97
Valeric acid	18.47 ^b^	20.42 ^b^	30.40 ^a^	2.78	0.03
Total SCFAs	734.66	719.02	750.86	42.83	0.87

^1^ Total SCFAs = acetic acid + propionic acid + isobutyric acid + butyric acid + isovaleric acid + valeric acid. ^2^ SEM, standard error of the mean (*n* = 6). ^a,b^ Different superscripts within a row indicate a significant difference (*p* < 0.05) or very significant difference (*p* < 0.01). NP, normal protein; LP, low protein; VLP, very low protein.

## Data Availability

The data presented in this study are available on request from the corresponding author.

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
