# Peer review of "Effects of Dietary Protein Restriction on Colonic Microbiota of Finishing Pigs"

_animals, 2022, doi:10.3390/ani13010009_

Round 1
Reviewer 1 Report
This article compares the gut microbial composition of finishing pigs on conventional and low protein diets, and finds some notable changes in the abundance of bacteria associated with protein degradation. However, there are still a few questions here that I would like to answer from you.
1. Line 14-16, there are some punctuation and initial case errors, please note and correct them.
2. Line 46-49, describe the establishment of gut microbes in finishing pigs and the basic changes. It is recommended to add the corresponding references here.
3. The trial focused on the changes and correlations of low protein diets on intestinal microorganisms and SCFAs, but it is necessary to provide phenotypic data such as body weight. If you have recorded the relevant data in the experiment, it would be better to provide them.
4. Some serum biochemical parameters were measured in the trial, indicating that you consider them necessary, and although the results did not have significant differences, the reasons for the lack of change in this section can also be briefly discussed in the discussion.
Author Response
Reviewer 1
This article compares the gut microbial composition of finishing pigs on conventional and low protein diets, and finds some notable changes in the abundance of bacteria associated with protein degradation. However, there are still a few questions here that I would like to answer from you.
- Line 14-16, there are some punctuation and initial case errors, please note and correct them.
Response: Thanks for your valuable comments, change has been made. Please refer to the lines14-16 for specific modifications.
- Line 46-49, describe the establishment of gut microbes in finishing pigs and the basic changes. It is recommended to add the corresponding references here.
Response: Thanks for your valuable comments,change has been made. Please refer to the lines49-52 and 305-308 for specific modifications.
- The trial focused on the changes and correlations of low protein diets on intestinal microorganisms and SCFAs, but it is necessary to provide phenotypic data such as body weight. If you have recorded the relevant data in the experiment, it would be better to provide them.
Response: Thanks for your valuable comments, I am sorry that I cannot provide complete growth performance data due to my personal reasons when feeding. And we believe that the focus of this article is to discuss the effect of low protein diet on colon microflora of finishing pigs.
- Some serum biochemical parameters were measured in the trial, indicating that you consider them necessary, and although the results did not have significant differences, the reasons for the lack of change in this section can also be briefly discussed in the discussion.
Response: Thanks for your valuable comments, we have added the discussion about the serum biochemical parameters. Please refer to the lines 222-231 for specific modifications.
Reviewer 2 Report
Overall the work needs to be reviewed for its grammar, some of the sentences do not make sense in their current form. E.g. Line 30 and 31 and 67 and 68 are good examples of this. There are several sections in the introduction that would benefit from further citation
Methods wise- I am unsure why the pigs needed to be slaughtered to get the data needed to prove/ disprove the hypothesis. This needs to be more clearly justified at this stage. Also line 69 highlights that pigs were killed after feeding, do you mean after day 38 of the feed trial or just straight after eating? how long after eating were they killed as would impact levels in blood.
Is 38 days long enough to see changes in the microbiota? Can this choice be backed up with literature.
Table 3 is well presented- the significant findings are easy to pull out
However I printed the file out in colour to edit and I could not see several on the numbers on figure 1 A- consider reviewing formatting to make it easier to read. same for figure 3 I cannot read any of those p values as it is too small.
Figure 2 is clear but the legend is the opposite way round with the colours than the figure making it confusing to read. fi that can be reformatted would be much better.
Discussion is suitable and has some good citations. Had a brief mention of the impact that the different level of fibre in the diets may have but think this would benefit from some wider consideration here.
Overall this appears to be an interesting study however I am not entirely sure of some of the methodological choices and further justification of these. Preferably backed up with some papers to show precedent in the choices would be beneficial.
Author Response
Overall the work needs to be reviewed for its grammar, some of the sentences do not make sense in their current form. E.g. Line 30 and 31 and 67 and 68 are good examples of this. There are several sections in the introduction that would benefit from further citation
Response: Thanks for your valuable comments, we have reviewed and revised the grammar of the whole article
Methods wise- I am unsure why the pigs needed to be slaughtered to get the data needed to prove/ disprove the hypothesis. This needs to be more clearly justified at this stage. Also line 69 highlights that pigs were killed after feeding, do you mean after day 38 of the feed trial or just straight after eating? how long after eating were they killed as would impact levels in blood.
Response: Thanks for your valuable comments, I am Sorry for any confusion. Because we need to measure the microbe and short-chain fatty acid content in colonic chyme, we need to slaughter pigs to obtain these. We have corrected the time error, Please refer to the lines 67-77 for specific modifications.
Is 38 days long enough to see changes in the microbiota? Can this choice be backed up with literature.
Response: Thanks for your valuable comments, We found it by searching the literature that 38 days is long enough to see changes in the microbiota. Chen et al found Eighteen barrows were randomly assigned to a normal (18%), low (15%), and extremely low (12%) dietary protein concentration group for 30 days. Colonic abundances of Ruminococcaceae, Christensenellaceae, Clostridiaceae_1, Spirochaetaceae, and Bacterodales_S24-7_group declined respectively, while proportions of Lachnospiraceae, Prevotellaceae, and Veillonellaceae increased with dietary protein reduction.
Zhu et al `s study showed that the abundance of unidentified Bacteria at the phylum level, and Halanaerobium and Butyricicoccusat at the genus level in the colonic digesta were significantly decreased by LP diet. And they just fed the pigs (100.38 ± 0.97 kg) for 28 days.
- Chen X, Song P, Fan P, et al. Moderate dietary protein restriction optimized gut microbiota and mucosal barrier in growing pig model[J]. Frontiers in cellular and infection microbiology, 2018, 8: 246.
- Zhu C, Yang J, Wu Q, et al. Low Protein Diet Improves Meat Quality and Modulates the Composition of Gut Microbiota in Finishing Pigs[J]. Frontiers in veterinary science, 2022, 9: 843957-843957.
Table 3 is well presented- the significant findings are easy to pull out
Response: Thanks for your valuable comments, I am sorry for that I did not understand clearly about this comments. We reexamined Table 3 and corrected the case of the shoulder markers indicating significance. Please refer to the Table 3. for specific modifications.
However I printed the file out in colour to edit and I could not see several on the numbers on figure 1 A- consider reviewing formatting to make it easier to read. same for figure 3 I cannot read any of those p values as it is too small.
Response: Thanks for your valuable comments, I'm really sorry for not processing the pictures well. We have adjusted the clarity of pictures 1 and 4. Please refer to the Figure 1 and Figure 4 for specific modifications.
Figure 2 is clear but the legend is the opposite way round with the colours than the figure making it confusing to read. fi that can be reformatted would be much better.
Response: Thanks for your valuable comments, the color of the legend is not the opposite of the figure, and we have reformatted figure 2.
Discussion is suitable and has some good citations. Had a brief mention of the impact that the different level of fibre in the diets may have but think this would benefit from some wider consideration here.
Response: Thanks for your valuable comments, We added some discussion in line 259-265.
Overall this appears to be an interesting study however I am not entirely sure of some of the methodological choices and further justification of these. Preferably backed up with some papers to show precedent in the choices would be beneficial.
Response: Thanks for your valuable comments, We reviewed the material methods section of the article and corrected its deficiencies.
Reviewer 3 Report
This manuscript mainly investigated the effect of dietary protein restriction on the colonic microbiota of finishing pigs. There are several interesting findings, however, there are some concerns about the manuscript. It seems that this experiment was conducted the same as the author’s previous study (see ref. 26 in the manuscript), the only difference is that the colonic microbiota was detected in this manuscript but it is cecal microbiota in the previous paper. However, the authors did not mention this in the manuscript, and did not compare the different results between colonic and cecal microbiota in the discussion. Moreover, there are many typos and grammar problems in the manuscript as follows:
1. In line 64, the sentence is not complete.
2. In line 62, the experiment period was 37 days, but in line 65 it indicated that the time was 38 days.
3. In line 114, there are no data for meat quality and FA composition.
4. In lines 175 and 181, it should be different protein diets, not different starch diets.
5. In fig.5, the title of the fig. is the concentration of colon biogenic amine content, however, they are VFA not amines. It also does not note the meaning of the stars.
6. In line 222, why it came out “ileal sample”?
7. In lines 226-227, missed “protein” after “dietary”.
8. In line 237, there are two “there”.
Author Response
This manuscript mainly investigated the effect of dietary protein restriction on the colonic microbiota of finishing pigs. There are several interesting findings, however, there are some concerns about the manuscript. It seems that this experiment was conducted the same as the author’s previous study (see ref. 26 in the manuscript), the only difference is that the colonic microbiota was detected in this manuscript but it is cecal microbiota in the previous paper. However, the authors did not mention this in the manuscript, and did not compare the different results between colonic and cecal microbiota in the discussion. Moreover, there are many typos and grammar problems in the manuscript as follows:
Response: Thanks for your valuable comments, we are very sorry for the trouble caused to you. This is a misquote and we have deleted it. Moreover, there are differences between the two articles. The previous article focused on the effect of low protein diet on meat quality, while this manuscript focuses on the effect of low protein diet on colon microbes and their metabolites.
- In line 64, the sentence is not complete.
Response: Thanks for your valuable comments, we have modified it. Please refer to the line67 for specific modifications.
- In line 62, the experiment period was 37 days, but in line 65 it indicated that the time was 38 days.
Response: Thanks for your valuable comments, I am sorry for the lack of clarity. We have reworked the language, Please refer to the line 67-75 for specific modifications.
- In line 114, there are no data for meat quality and FA composition.
Response: Thanks for your valuable comments, I am sorry for this mistake, and we have deleted it. Please refer to the line 123 for specific modifications.
- In lines 175 and 181, it should be different protein diets, not different starch diets.
Response: Thanks for your valuable comments, we have modified it. Please refer to the line 186 and 192 for specific modifications.
- In fig.5, the title of the fig. is the concentration of colon biogenic amine content, however, they are VFA not amines. It also does not note the meaning of the stars.
Response: Thanks for your valuable comments, we have modified it. Please refer to the line 220 and 222-223 for specific modifications.
- In line 222, why it came out “ileal sample”?
Response: Thanks for your valuable comments, it is colon samples, and we have modified it. Please refer to the line 243 for specific modifications.
- In lines 226-227, missed “protein” after “dietary”.
Response: Thanks for your valuable comments, we have modified it. Please refer to the line 248 for specific modifications.
- In line 237, there are two “there”.
Response: Thanks for your valuable comments, we have deleted it. Please refer to the line 259 for specific modifications.
Round 2
Reviewer 3 Report
The authors answered all my concerns. Please delete "cecum" in line 139.